# Influence of “Face-to-Face Contact” and “Non-Face-to-Face Contact” on the Subsequent Decline in Self-Rated Health and Mental Health Status of Young, Middle-Aged, and Older Japanese Adults: A Two-Year Prospective Study

**DOI:** 10.3390/ijerph19042218

**Published:** 2022-02-16

**Authors:** Yoshinori Fujiwara, Kumiko Nonaka, Masataka Kuraoka, Yoh Murayama, Sachiko Murayama, Yuta Nemoto, Motoki Tanaka, Hiroko Matsunaga, Koji Fujita, Hiroshi Murayama, Erika Kobayashi

**Affiliations:** Research Team for Social Participation and Community Health, Tokyo Metropolitan Institute of Gerontology, Tokyo 173-0015, Japan; nonaka@tmig.or.jp (K.N.); mkuraoka@tmig.or.jp (M.K.); yhoyho05@tmig.or.jp (Y.M.); sachikom@tmig.or.jp (S.M.); nemoto@tmig.or.jp (Y.N.); moto15@tmig.or.jp (M.T.); hiroko_m@tmig.or.jp (H.M.); fujita@tmig.or.jp (K.F.); murayama@tmig.or.jp (H.M.); erikak@tmig.or.jp (E.K.)

**Keywords:** social contact, face-to-face contacts, non-face-to-face contacts, age classes, self-rated health, mental health, longitudinal study

## Abstract

This study aims to identify the independent influence of face-to-face contact (FFC) and non-face-to-face contact (NFFC) on the subsequent decline in self-rated health and mental health status by age. A total of 12,000 participants were randomly selected among residents in the study area, and 1751 of them responded to both the 2016 and 2018 mail surveys. The participants were subsequently classified into three age groups (25–49: Young adults; 50–64: Mid-aged adults; and 65–84: Older adults). Social contact was assessed by computing the frequencies of FFC and NFFC. Multiple logistic regression analysis showed the risk of social contact on the decline in self-rated health and World Health Organization-Five Well-Being Index. Both FFC and NFFC were significantly associated with maintaining mental health; however, the impacts of FFC on mental health were more significant than that of NFFC among older adults and young adults. Compared with the no contact group, FFC was significantly associated with maintaining self-rated health in mid-aged adults. The influence of FFC and NFFC on health differed by age group.

## 1. Introduction

Social contact is an essential behavior in the daily lives of people, which can affect individuals’ health status. A general lack of social contact and a reduced social network size is defined as social isolation. A previous systematic review revealed that social isolation is associated with general and mental health problems, such as anxiety and depression, and is one of the causes of mortality in adults [1]. Additionally, a meta-analysis from a review article showed that individuals who are more firmly embedded in their social surroundings are healthier than those with relatively weak social ties, an indication that the risk effect of social contact on human lives exceeds the risk effect of other lifestyle factors such as smoking, drinking, and exercise habits [2].

Social isolation is a common issue for all ages in Japan. Severe isolation-related phenomena such as “Koritsushi”, which means isolated or lonely death [3], and “Hikikomori”, a severe form of social withdrawal [4,5], may frequently occur across all ages in Japan. Therefore, to avoid social isolation, the Japanese government has urged citizens not to loose traditional family-based social contact. Moreover, as a national policy, Japan is the second country to appoint a “minister of loneliness” in 2021, following Britain in 2018.

Although several studies have been conducted on social isolation, age-specific evidence related to social isolation and health problems is currently underexplored in Japan and other countries. The participants in most previous studies are elderly, and only a few studies have focused on young and middle-aged adults [6,7,8]. Consequently, it is essential to explore all age categories while examining relationships between social contact and health-related problems to propose strategies to cope with social isolation.

Social isolation is defined in this study as low frequency of contact with others; this definition discriminates social isolation from feelings of isolation or loneliness, in accordance with Townsend [9]. The frequency of contact with non-residential family members, relatives, friends, and neighbors was measured to define social isolation operationally. Based on the definition, previous studies have referred to people who have contact with others less than once a week as “isolated adults” [10]. These studies found that isolated status predicted adverse health outcomes [11,12,13].

However, most previous studies do not distinguish between face-to-face contact (FFC) and non-face-to-face contact (NFFC) with others [11,12,13,14,15].

It is insufficient to discuss social isolation without considering the type of contact. This is because social isolation has become prevalent globally since the emergence of the 2019 coronavirus (COVID-19) pandemic. Keeping social distance and refraining from gathering or meeting are measures to prevent contracting COVID-19. These preventive measures have decreased the frequency of FFC with others, compared with NFFC, such as using telephone, email, social networking services (SNS), and facsimile for communication. NFFC seems to have a more significant role in people’s daily lives because it may complement FFC as an alternative means of communication in the aftermath of the COVID-19 pandemic. Nevertheless, it remains unclear which type of social contact (FFC or NFFC) has a more significant impact on health by age group.

This is the first study to explore and identify the independent influence of FFC and NFFC, compared to no contact (NC), on the subsequent decline in self-rated health and mental health status among Japanese adults based on age group.

## 2. Materials and Methods

### 2.1. Study Area and Participants

This study used survey data from a two-year longitudinal study of adults aged 25–84 years living in Kita ward, the northern part of central Tokyo, and the Tama ward of Kawasaki city. The Tama ward is a typical commuter city in the western suburb of the Tokyo metropolitan area in Japan. In 2016, the total populations of the Kita and Tama wards were 340,000 and 210,000, respectively. In Kita, 25% of the population was over 65 years (older population) in 2016, while in Tama, it was 19%. Baseline data (T1) for this study were collected in 2016, and follow-up data were collected in 2018 (T2).

Figure 1 shows that the baseline mailed questionnaires were randomly distributed to 12,000 residents in the Kita ward (*n* = 6000) and the Tama ward (*n* = 6000). In the Kita ward and Tama ward, the ratio of extracted subjects was 1:1:2 for 65–84 years, 50–64 years, and 25–49 years, respectively. Of the 3963 participants (Kita ward: 1938; Tama ward: 2025) who responded to the baseline survey, 1824 responded to the follow-up survey in July 2018 (see Figure 1 for details). We analyzed data from the 1751 participants who completed the questionnaire. The study was conducted in accordance with the Declaration of Helsinki, and approved by the Ethics Committee of the Tokyo Metropolitan Institute of Gerontology (protocol code 28KEN-1042; date of approval: 1 June 2016).

### 2.2. Measures and Analysis Strategy

The questionnaires consisted of health-related factors, demographic, lifestyle-related, and psychosocial variables.

In terms of significant outcome variables, the health-related factors included psychological health status and self-rated health. For self-rated health, respondents rated their health as very good, good, poor, or very poor. The self-rated scales were combined into a dichotomous variable, “poor” consisting of “very poor” and “poor”, and “good” consisting of “good” and “very good”. Psychological well-being was evaluated using the World Health Organization-Five Well-Being Index (WHO-5). WHO-5 is a sensitive and specific depression screening tool with very high applicability across study fields [16]. The WHO-5 scale contains five items, all positively worded. It is among the most widely used questionnaires for assessing subjective psychological well-being [17]. A maximum score of “25” indicates optimal well-being, whereas a “0” indicates minimal well-being. We coded WHO-5 scores as a binary variable with a standard cut-off criterion of 12/13. That is, scores of 12 or less were regarded as a poor condition.

The primary independent variable from the T1 data was social contact in this study. It was measured by (1) frequency of FFC with non-residential family, relatives, friends, and neighbors; and (2) frequency of NFFC (e.g., via telephone, email, letters) with non-residential family and relatives, friends, and neighbors. The frequencies of FFC and NFFC were measured as follows: “every day”, “four to five times a week”, “two to three times a week”, “once a week”, “two to three times a month”, “once a month”, “less than once a month”, and “NC”. In a previous study [18], social contact status was categorized into “having face-to-face contact with or without non-face-to-face contact” (FFC), “non-face-to-face contact only” (NFFC), and “no contact (i.e., isolation)” (NC) based on whether respondents had contact at least once a week with anyone, including kin living apart, friends, and neighbors.

The demographic variables included age, gender, years of education, living arrangements, years of residence, and financial status. The living arrangement consists of “living alone” vs. “living with others”. In terms of financial status, participants were asked to respond to their financial situation with four options, ranging from having substantial financial leeway to having a very little leeway. Participants financial statuses were then classified as good (I have above average financial leeway) or poor (My finances are tight to very tight). We also asked the participants about their history of chronic conditions such as stroke, heart disease, cancer, and diabetes.

### 2.3. Statistical Methods

All data were analyzed using the SPSS/PC+ Statistical Software for Windows version 20.0.

Participants were stratified into three age groups: Group O, older adults (65–84 years); Group M, mid-aged adults (50–64 years); and Group Y, young adults (25–49 years). First, to analyze the baseline characteristics of the participants who responded to both T1 and T2, stratified by the three age groups (Groups O, M, and Y), a Chi-squared test was used for categorical variables and analysis of variance (ANOVA) for continuous variables.

Second, to clarify the cross-sectional relationship between FFC, NFCC, and NC and self-rated health and mental health, we used multiple logistic regression, adjusted for conventional confounding variables analyses by age group at baseline.

Third, to identify the impacts of social contact on declining self-rated health and mental health during the two-year follow-up period (T1–T2), multiple logistic regression analyses were performed. The analysis included participants who had good self-rated health and good WHO-5 (≥13) at the baseline, and the confounding variables were entered into the models and were mandatory after we considered the degree of multiple collinearities.

## 3. Results

Table 1 presents the characteristics of those who responded to baseline and follow-up surveys stratified by age group. Group O was significantly less isolated and more mentally healthy in the WHO-5 index than Groups M and Y. The proportion of those who had significantly poor self-rated health was greater among Group M, compared to Group Y and Group O.

Although the data are not shown in Table 1, female participants had more FFCs and less NC compared to male participants (52.1% vs. 32.4%; 22.5% vs. 46.2%), respectively (*p* < 0.001). We found the same trends among Groups O, M, and Y.

Table 2 shows the results of the cross-sectional multiple logistic regression analyses by age group at the baseline survey. For Groups M and Y, FFC, compared with isolation, was negatively associated with poor self-rated health, even after adjustment for age, gender, history of chronic conditions, years of education, living arrangement, and subjective economic condition (OR = 0.50, 95% CI: 0.26–0.94; OR = 0.40, 95% CI: 0.19–0.83, respectively). Having both NFFC and FFC was also significantly associated with poor mental health, as assessed by the WHO-5 for isolation in all age groups. The odds ratios of having FFC were more significant than those of NFFC.

Table 3 displays the results of longitudinal multiple logistic regression analyses by age group among participants who had good self-rated health or good WHO-5 scores. For Group M, FFC, compared with isolation, was a protective predictor of the subsequent decline in self-rated health, even after adjustment for the same confounding factors as in Table 2 (OR = 0.28, 95% CI: 0.10–0.80). Having both NFFC and FFC—in comparison with isolation—indicated a subsequent decline in poor mental health status for Group O (OR = 0.45, 95% CI: 0.21–0.97; OR = 0.27, 95% CI: 0.14–0.51) and Group Y (OR = 0.47, 95% CI: 0.25–0.88; OR = 0.42, 95% CI: 0.23–0.74). The odds ratios of having FFC were stronger than those of NFFC for Groups O and Y.

## 4. Discussion

This two-year prospective study aimed to examine the independent influence of “having FFC” or “having NFFC only” compared to “NC” on the subsequent decline in self-rated health and mental health status of different age groups over two years. Our findings revealed that the associations of FFC and NFFC social contacts with self-rated health and mental health were different between Groups O, M, and Y, respectively.

### 4.1. Cross-Sectional and Longitudinal Association of Social Contact with Self-Rated Health

The present study demonstrated positive relationships between FFC and good self-rated health among Groups M and Y.

A previous review article demonstrated the effects of social networks on health-related behaviors which was strongly associated with self-rated health for young-aged participants whose mean age was 32.4 years. There were significant intervention effects for their health outcomes, including alcohol misuse, well-being, change in HbA1c, and smoking cessation [19]. They might be supported to maintain or improve health-related behaviors by their family members, friends, or colleagues with FFC.

For example, social support and social networks were associated with promoting physical activity [20,21]. The National Health Interview Survey (NHIS) showed that participants who had the highest level of social integration (both calling and seeing) with friends or family members in the past two weeks had more odds of being sufficiently physically active than those who reported no contact with friends and family [22]. However, while friend integration predicted markedly greater odds of physical activity compared to a moderate level of social integration (either seeing or calling), greater family integration showed a decrease in physical activity.

A previous study reported that the frequency of FFC with family remains relatively stable across the life course. However, the frequency of visits with non-family members declines [23]. These relationships between self-rated health and FFC were slightly attenuated in Group O. Although we measured social contact, combining family members and non-family members, our findings were supported by the previous studies mentioned above.

There was a likelihood of maintaining or improving self-rated health. Furthermore, having FFC compared with NC further protected the subsequent self-rated health decline in Group M, although we did not search frequency or contents of FFC by family or relatives in detail. On the other hand, the history of chronic conditions strongly influenced self-rated health rather than FFC or NFFC with others in Group O.

### 4.2. Cross-Sectional and Longitudinal Association of Social Contact with Mental Health

Conventionally, there is evidence that social contacts are protective factors for good mental health [24], especially among older adults [25]. However, little is known about the impact of social contact or social isolation on mental health across different age groups [6,26]. Regression coefficients or odds ratios showed that social connection improved mental health across age groups from young to older adults [6,26].

The current study showed that FFC and NFFC were significantly associated with maintaining mental health, and the impact of FFC was more significant than that of NFFC in cross-sectional and longitudinal analyses among Groups O and Y. This difference in the present study may be caused by the influence of NFFC tools, which might make NFFC less effective for mental health compared to FFC.

Although the proportion of those who were isolated increased with age in previous studies [6], our study demonstrated that Group O was less isolated compared to Groups M and Y. This may partially be because Japanese older adults appreciate formal and informal support to prevent social isolation and functional decline, including encouragement to participate in various social activities and community integrated care systems that provide assistance with daily activities and safety-related support [27]. As Japan is at a mid-level for lack of social support across all age groups [28], the number of isolated young adults and middle-aged adults did not consistently differ from that of the older adults. The prevalence of suicide induced by isolation among middle-aged adults was similar to that of older adults [29].

### 4.3. Limitations and Strength

First, the present study did not discuss SNS type and usage in detail, even though NFFC includes SNS as well as telephone, email, letters, and so on. Generally speaking, the significant impact of SNSs on the relationship between health and NFFC may be due to online social interactions, especially among adolescents and adults. SNSs seem to be a helpful tool for communicating and sharing emotions and thoughts with others. Several studies have demonstrated the merits of using SNS in social relationships [30,31], leading to good health. Conversely, other studies have indicated that SNS induced stressful experiences, with users showing poor well-being [32,33].

A previous study reported the differences in SNS type and usage by age group to clarify the positive and negative associations between SNS usage and health. Frequent posting on Facebook was associated with better well-being among middle-aged adults. Young adults who frequently visited Instagram showed a tendency toward better well-being and lower distress symptoms. On the contrary, frequent usage of Twitter was associated with distress symptoms or feelings of loneliness across all generations [34].

Although the current study did not examine the type and usage of SNS in detail, both FFC and NFFC were significantly associated with maintaining mental health, and the impact of FFC on mental health was more significant than that of NFFC in cross-sectional and longitudinal analyses among Groups O and Y. The influence of SNS on NFFC in the present study might not be very strong or excessively harmful to mental health.

Moreover, due to the COVID-19 pandemic, we are progressing to online communication (OLC) in all of our daily life activities, including business and private lives. Since we can see each other through screens when communicating online, we might regard OLC as an intermediate between FFC and NFFC. However, the present questionnaire did not include OLC-related items. Future studies should determine the influence of OLC on health in comparison with FFC and NFFC.

Second, our findings could not clarify the range of friendships because junior and senior colleagues in participants’ workplaces were excluded as contact subjects and, therefore, were not regarded as friends in the questionnaire. However, the impact of workplace associates may become weaker as a result of drastic changes in the work system—the shift to telework as a result of COVID-19—in addition to gradual changes in work-life balance.

Third, the participants were recruited from an urban city. Previous studies have observed that sociocultural factors are critical components associated with maintaining functional capacities among older adults [35]. Kita City is a downtown area of Tokyo, and Kawasaki city is a typical suburban city near central Tokyo, where the lifestyles of residents are rather heterogeneous, consisting of those who moved from other prefectures of Japan since the 1970s and traditional settlers. Future studies are needed to demonstrate the findings of this study in different socio-cultural perspectives, including rural or downtown areas.

Despite these limitations, this study has considerable strengths and has filled a noticeable gap. This study is the first to examine the causal relationship between the impact of social contacts, classified by FFC vs. NFFC, on health by age group. The findings may indicate models of mechanism between social contact and health status.

Cohen proposed two general theoretical models of the mechanism by which social contacts might influence health: the stress-buffering and main effects models [36]. The buffering hypothesis suggests that social relationships may provide resources that promote adaptive behavioral or neuroendocrine responses to acute or chronic stressors. The main effects model proposes that social relationships may be associated with protective health effects through more direct means, such as cognitive, emotional, behavioral, and biological influences not explicitly intended as help or support. In the future, researchers should examine the two theoretical models of the mechanism between health and social relationships with FFC and NFFC by age group.

## 5. Conclusions

The findings suggest that having both FFC and NFFC were significantly associated with maintaining good mental health, and the impacts of FFC on mental health were more significant than NFFC in both cross-sectional and longitudinal analyses among Groups O and Y. However, FFC, compared with NC, was significantly associated with maintaining good self-rated health in Group M, both cross-sectionally and longitudinally. Therefore, the influence of FFC and NFFC on self-rated health and mental health differ by age group.

## Figures and Tables

**Figure 1 ijerph-19-02218-f001:**
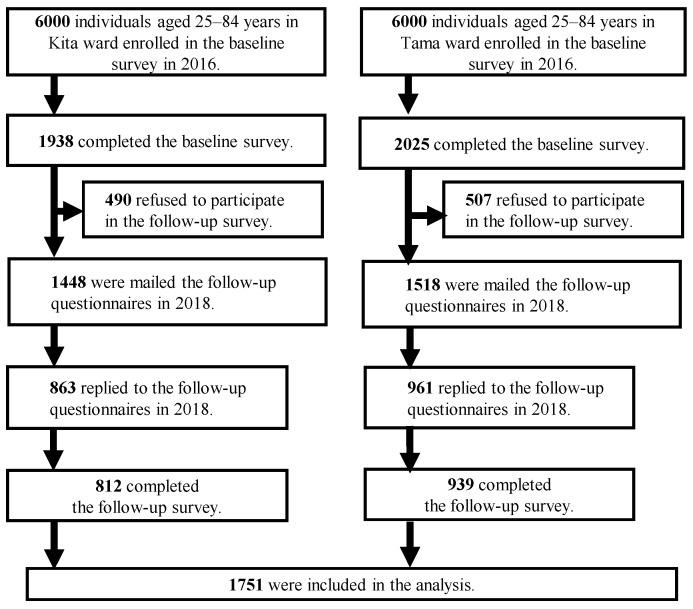
Flow chart of this study.

**Table 1 ijerph-19-02218-t001:** Distributions of study participant characteristics stratified by age groups at baseline.

		(1) Group O: 65–84 Years	(2) Group M: 50–64 Years	(3) Group Y: 25–49 Years	Total	*p* ^a^	Multiple Comparison Procedure		
		*n* = 649	*n* = 533	*n* = 569	*n* = 1751		(1) vs. (2)	(2) vs. (3)	(3) vs. (1)
Age	Mean (SD)	72.4 (5.3)	57.2 (4.4)	39.4 (6.7)	57.0 (14.8)	<0.001	<0.001	<0.001	<0.001
Field	Tama ward	54.1%	54.2%	52.5%	53.6%	0.820	-	-	-
Gender	female	54.4%	63.4%	67.7%	61.5%	<0.001	0.002	0.145	<0.001
Social contact status ^b^	NC	26.1%	37.3%	37.9%	33.4%	<0.001	<0.001	0.792	<0.001
	NFFC	18.2%	27.0%	28.2%	24.2%				
	FFC	55.7%	35.8%	33.9%	42.4%				
WHO-5 ^c^ (range: 0–25)	poor (<13)	25.4%	32.8%	33.6%	30.3%	0.003	0.006	0.798	0.002
Self-rated health ^d^	poor	15.5%	16.4%	11.5%	14.5%	0.043	0.688	0.018	0.044
Number of history of chronic conditions	≥1	34.9%	16.7%	7.2%	20.4%	<0.001	<0.001	<0.001	<0.001
Years of education	<13	58.4%	30.5%	17.3%	36.5%	<0.001	<0.001	<0.001	<0.001
Living arrangement	living alone	22.7%	10.6%	18.1%	17.5%	<0.001	<0.001	0.001	0.058
Subjective financial Situation ^e^	not so good	21.2%	25.1%	22.8%	22.9%	0.282	-	-	-

Note: SD = standard deviation. ^a^
*p*-values were evaluated using χ^2^ test for categorical variables and analysis of variance for continuous variables. ^b^ NC = “no contact (i.e., isolation)”; FFC = “face-to-face contact and/or non-face-to-face contact”; NFFC = “non-face-to-face contact only”. ^c^ World Health Organization-Five Well-Being Index (WHO-5). ^d^ Self-rated is combined into a dichotomous variable: “poor”, which consists of “very poor” and “poor”, and “good”, which consists of “good” and “very good”. ^e^ Subjective financial situation is classified as good (I have above average financial leeway) or poor (My finances are tight to very tight).

**Table 2 ijerph-19-02218-t002:** Multiple logistic regression estimating the odds of poor self-rated health ^a^ and WHO-5 ^b^ at baseline.

Poor Self-Rated Health ^a^	Reference	Category	(1) Group O: 65–84 Years			(2) Group M: 50–64 Years			(3) Group Y: 25–49 Years		
			OR	95% C.I.	*p*	OR	95% C.I.	*p*	OR	95% C.I.	*p*
Field	Kita ward	Tama ward	0.94	0.74–1.19	0.594	1.13	0.88–1.46	0.327	1.08	0.81–1.43	0.619
Gender	male	female	1.65	0.98–2.78	0.061	1.83	1.01–3.30	0.045	1.18	0.63–2.20	0.608
Age	1 year crement		1.04	1.00–1.09	0.072	0.99	0.94–1.05	0.775	1.01	0.97–1.06	0.575
Social contact status ^c^	NC	NFFC	1.04	0.52–2.06	0.923	0.87	0.47–1.63	0.662	1.02	0.53–1.97	0.963
		FFC	0.68	0.38–1.22	0.197	0.50	0.26–0.94	0.031	0.40	0.19–0.83	0.014
Number of history of chronic conditions	0	≥1	5.42	3.29–8.93	0.000	4.39	2.49–7.73	0.000	6.56	3.03–14.18	0.000
Years of education	<13	≥13	0.77	0.45–1.29	0.314	0.85	0.50–1.44	0.542	0.70	0.35–1.40	0.317
Living arrangement	living with someone	living alone	1.18	0.68–2.05	0.563	2.01	0.99–4.10	0.055	1.32	0.64–2.72	0.453
Subjective financial situation ^d^	good	poor	2.22	1.29–3.83	0.004	2.29	1.35–3.88	0.002	3.19	1.77–5.73	0.000
Poor WHO-5 ^b^ (<13 scores)	Reference	Category	Group O: 65–84 years			Group M: 50–64 years			Group Y: 25–49 years		
			OR	95% C.I.	*p*	OR	95% C.I.	*p*	OR	95% C.I.	*p*
Field	Kita ward	Tama ward	0.83	0.67–1.02	0.072	1.12	0.92–1.36	0.270	0.90	0.75–1.08	0.249
Gender	male	female	1.15	0.73–1.80	0.547	0.95	0.62–1.46	0.812	1.03	0.69–1.53	0.901
Age	1 year crement		0.97	0.93–1.00	0.075	0.96	0.92–1.00	0.075	1.01	0.98–1.04	0.405
Social contact status ^c^	NC	NFFC	0.55	0.31–0.96	0.036	0.56	0.34–0.93	0.025	0.63	0.41–0.99	0.043
		FFC	0.21	0.13–0.35	0.000	0.50	0.31–0.81	0.005	0.30	0.19–0.47	0.000
Number of history of chronic conditions	0	≥1	1.38	0.90–2.11	0.143	1.54	0.92–2.60	0.103	0.84	0.40–1.76	0.635
Years of education	<13	≥13	0.60	0.38–0.93	0.023	0.70	0.46–1.07	0.102	1.26	0.76–2.08	0.378
Living arrangement	living with someone	living alone	1.19	0.73–1.92	0.491	1.54	0.83–2.87	0.171	1.55	0.97–2.49	0.068
Subjective financial situation ^d^	good	poor	2.48	1.56–3.94	0.000	2.10	1.36–3.26	0.001	1.93	1.26–2.96	0.003

Note: OR = odds ratio; C.I. = confidence interval. ^a^ Self-rated is combined into a dichotomous variable: “poor”, which consists of “very poor” and “poor”, and “good”, which consists of “good” and “very good”. ^b^ World Health Organization-Five Well-Being Index (WHO-5). ^c^ NC = “no contact (i.e., isolation)”; FFC = “having face-to-face contact and/or non-face-to-face contact”; NFFC = “non-face-to-face contact only”. ^d^ Subjective financial situation is classified as good (I have above average financial leeway) or poor (My finances are tight to very tight).

**Table 3 ijerph-19-02218-t003:** Multiple logistic regression estimating the odds of subsequent decline in self-rated health ^a^ and WHO-5 ^b^ between baseline and follow-up among subjects who had good self-rated health or good WHO-5 scores at baseline.

Decline in Self-Rated Health ^a^	Reference	Category	Group O: 65–84 Years			Group M: 50–64 Years			Group Y: 25–49 Years		
			OR	95% C.I.	*p*	OR	95% C.I.	*p*	OR	95% C.I.	*p*
Field	Kita-ward	Tama-ward	0.71	0.36–1.37	0.304	0.42	0.18–0.99	0.048	0.68	0.31–1.53	0.351
Gender	male	female	0.78	0.38–1.58	0.485	0.79	0.33–1.88	0.587	0.45	0.20–1.02	0.057
Age	1 year crement		1.02	0.96–1.09	0.473	1.06	0.96–1.17	0.259	0.87	0.82–0.93	0.000
Social contact status ^c^	NC	NFFC	0.53	0.19–1.46	0.221	0.47	0.17–1.34	0.159	0.75	0.28–2.02	0.573
		FFC	0.64	0.30–1.36	0.240	0.28	0.10–0.80	0.017	0.79	0.31–2.01	0.624
Number of history of chronic conditions	0	≥1	2.71	1.37–5.34	0.004	2.43	0.93–6.39	0.071	3.01	0.60–15.26	0.183
Years of education	<13	≥13	0.67	0.33–1.35	0.258	0.38	0.17–0.87	0.021	0.28	0.11–0.75	0.011
Living arrangement	living with someone	living alone	0.96	0.43–2.14	0.911	0.17	0.02–1.44	0.104	0.84	0.31–2.31	0.739
Subjective financial situation ^d^	good	poor	2.86	1.38–5.91	0.005	3.21	1.39–7.38	0.006	1.79	0.76–4.20	0.181
Poor WHO-5 ^b^ (<13 scores)	Reference	Category	Group O: 65–84 years			Group M: 50–64 years			Group Y: 25–49 years		
			OR	95% C.I.	*p*	OR	95% C.I.	*p*	OR	95% C.I.	*p*
Field	Kita-ward	Tama-ward	0.53	0.31–0.90	0.019	1.21	0.71–2.06	0.487	0.88	0.54–1.44	0.613
Gender	male	female	0.53	0.29–0.96	0.037	1.46	0.80–2.69	0.220	0.65	0.39–1.09	0.100
Age	1 year crement		1.00	0.95–1.05	0.908	1.00	0.94–1.07	0.914	0.99	0.96–1.03	0.756
Social contact status ^c^	NC	NFFC	0.45	0.21–0.97	0.043	0.81	0.42–1.57	0.536	0.47	0.25–0.88	0.019
		FFC	0.27	0.14–0.51	0.000	0.66	0.34–1.30	0.227	0.42	0.23–0.74	0.003
Number of history of chronic conditions	0	≥1	1.29	0.73–2.27	0.384	1.39	0.67–2.85	0.376	3.13	1.35–7.25	0.008
Years of education	<13	≥13	0.75	0.43–1.30	0.298	0.83	0.46–1.50	0.541	0.95	0.50–1.82	0.880
Living arrangement	living with someone	living alone	0.72	0.37–1.39	0.325	2.36	1.03–5.39	0.043	1.38	0.73–2.62	0.324
Subjective financial situation ^d^	good	poor	2.76	1.48–5.12	0.001	1.42	0.74–2.71	0.295	2.37	1.33–4.20	0.003

Note: OR = odds ratio; C.I. = confidence interval. ^a^ Self-rated is combined into a dichotomous variable: “poor”, which consists of “very poor” and “poor”, and “good”, which consists of “good” and “very good”. ^b^ World Health Organization-Five Well-Being Index (WHO-5). ^c^ NC = “no contact (i.e., isolation)”; FFC = “having face-to-face contact and/or non-face-to-face contact”; NFFC = “non-face-to-face contact only”. ^d^ Subjective financial situation is classified as good (I have above average financial leeway) or poor (My finances are tight to very tight).

## Data Availability

The data presented in this study are available on request from the corresponding author.

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
