# Peer review of "Influence of “Face-to-Face Contact” and “Non-Face-to-Face Contact” on the Subsequent Decline in Self-Rated Health and Mental Health Status of Young, Middle-Aged, and Older Japanese Adults: A Two-Year Prospective Study"

_ijerph, 2022, doi:10.3390/ijerph19042218_

Round 1

Reviewer 1 Report

All tables and the study flow chart are clear and very helpful. The article can benefit from few clarifications.

(i) The authors used an extensive set of measures to examine the background and lifestyle of different age groups. Were any self-reported depression screenings used in the current study? Depression is an important factor to consider when measuring one's social isolation. If no depression screenings were used, it is going to be helpful to include that as one of the study limitations.

(ii) Did the authors see any within-group differences for older adults who lived alone versus those living with a spouse or other family members? If so, some of those details would be helpful to further explain the reported higher well-being score among the older adults.

(iii) This is a very interesting study that coincides with the ongoing pandemic that has challenged all of us in terms of social isolation and extreme dependence of social media apps and other digital means of communication. One future direction would be to follow-up with these participants now in the COVID times to see how their FFC, NFFC, and NC ratings have changed.

(iv) It will be also helpful to add few statements about any gender differences that may been observed with regard to frequency of FFCs, NFFCs, NCs, and use of social medial.

Author Response

Reviewer  1’ s comments and suggestions for authors

All tables and the study flow chart are clear and very helpful. The article can benefit from few clarifications.

Answer: We appreciate the comments by Reviewer 1. We have revised the paragraphs according to the reviewer’s suggestions.

Q(i) The authors used an extensive set of measures to examine the background and lifestyle of different age groups. Were any self-reported depression screenings used in the current study? Depression is an important factor to consider when measuring one's social isolation. If no depression screenings were used, it is going to be helpful to include that as one of the study limitations.

Answer: We used the WHO-5 as a self-reported depression screening scale. A review* demonstrated that the WHO-5 has high clinimetric validity. In addition, it showed that the scale can be used as an outcome measure balancing the wanted and unwanted effects of treatments, is a sensitive and specific screening tool for depression, and its applicability across study fields is very high. We added the reference article shown below (Topp et al., 2015) according to the reviewer’s comments.

REFERENCE

*Topp, C.; Østergaard, S.; Søndergaard, S.; Bech, P. The WHO-5 Well-Being Index: A systematic review of the literature. Psychother Psychosom 2015, 84, 167–76. 10.1159/000376585.

Q(ii) Did the authors see any within-group differences for older adults who lived alone versus those living with a spouse or other family members? If so, some of those details would be helpful to further explain the reported higher well-being score among the older adults.

Answer: I am afraid that we did not ask about family members in detail in the present study.

Q(iii) This is a very interesting study that coincides with the ongoing pandemic that has challenged all of us in terms of social isolation and extreme dependence of social media apps and other digital means of communication. One future direction would be to follow-up with these participants now in the COVID times to see how their FFC, NFFC, and NC ratings have changed.

Answer: Thank you for your important suggestion. We would like to try to follow-up this in the future.

(iv) It will be also helpful to add few statements about any gender differences that may been observed with regard to frequency of FFCs, NFFCs, NCs, and use of social media.

Answer: The proportions of FFCs, NFFCs, and NCs for male and female participants were 32.4%, 21.4%, 46.2% vs. 52.1% 25.4%, 22.5%, respectively (p<.001). I am afraid that I did not ask about using social media in detail. 

To clarify this, I have added an explanation to the revised text (Line 167-Line 169 in the revised text):

Although the data are not shown in Table 1, female participants had more FFCs and less NC compared to male participants (52.1% vs. 32.4%; 22.5% vs. 46.2%), respectively (p<.001). We found the same trends among Groups O, M, and Y. 

Reviewer 2 Report

Overall, I found the manuscript to be concise, well-written and informative. Well done to the authors. I have just a few more specific comments to be addressed to help improve the manuscript in a couple of areas.

Did you have any specific hypotheses that you could include at the end of the introduction? Or are you arguing that the research is purely exploratory?

“Figure 1 shows that the baseline mailed questionnaires were randomly distributed to 85 16,620 residents in the Kita ward (n=6,000) and the Tama ward (n=6,000).” This sentence confused me. The two n=6000 do not match up with the 16,620. Can you clarify what these numbers are referring to?

Figure 1 is very useful for understanding the study design ?

As I understand, social contact is assessed using subjective measures (i.e. participants self-report how often they have FFC and NFFC). But on page 2 you have said the study is defining social isolation as objective flow frequency of contact… I understand that by objective in this case you mean reported frequency of interactions rather than just how isolated people feel, but I worry that the use of the term objective may be troublesome because the measure of social contact is still subjectively reported (i.e., not externally validated, people can lie).

I wasn’t completing sure on why logistic rather than linear regression was used in some cases. I.e. what was the value is coding WHO-5 scores as a binary variable over just looking at them as a linear scale. A sentence or two to explain and justify this approach might be useful to include in the ‘materials and methods’ section.

There were some points in the discussion where I wasn’t clear if you were referring to your own findings or that of previous research. For example, on line 230 you say “The trajectories of social contact indicated differences between contact with family 230 members and non-family members; the frequency of social contact with family or relatives 231 increased, while that with friends or neighbors decreased.” But I couldn’t recall you talking about this finding in the results section…

Line 253, you say “influence of SNS…” but I don’t think you had told the reader prior to this what SNS referred to?

I did not see the value in having the ‘Measures and analysis strategy’ heading within the discussion as most of the content that came under it was related to the results of the research.

Author Response

Overall, I found the manuscript to be concise, well-written and informative. Well done to the authors. I have just a few more specific comments to be addressed to help improve the manuscript in a couple of areas.

Answer: We appreciate the comments by Reviewer 2. We have revised the paragraphs so they are less wordy with sharper text.

Q 1. Did you have any specific hypotheses that you could include at the end of the introduction? Or are you arguing that the research is purely exploratory?

Answer: We regard this research as purely exploratory. Therefore, we revised the text accordingly. The text now reads (Lines 74–76 in the revised text): “This is the first study to explore and identify the independent influence of FFC and NFFC, compared to  no contact (NC), on the subsequent decline in self-rated health and mental health status among Japanese adults based on age group.”

Q 2. “Figure 1 shows that the baseline mailed questionnaires were randomly distributed to 16,620 residents in the Kita ward (n=6,000) and the Tama ward (n=6,000).” This sentence confused me. The two n=6000 do not match up with the 16,620. Can you clarify what these numbers are referring to?  Figure 1 is very useful for understanding the study design ?

Answer: I am very sorry for my misprint. We revised the text accordingly. The text now reads (Line 87 in the revised text): “Figure 1 shows that the baseline mailed questionnaires were randomly distributed to 12,000 residents in the Kita ward (n=6,000) and the Tama ward (n=6,000).” Similarly, we have revised the ABSTRACT.

Q 3 . As I understand, social contact is assessed using subjective measures (i.e. participants self-report how often they have FFC and NFFC). But on page 2 you have said the study is defining social isolation as objective flow frequency of contact… I understand that by objective in this case you mean reported frequency of interactions rather than just how isolated people feel, but I worry that the use of the term objective may be troublesome because the measure of social contact is still subjectively reported (i.e., not externally validated, people can lie).

Answer: According to the reviewer’s comment, we now abstain from using “objective” and “subjective” in the text. The text now reads (Line 53–Line 55 in the revised text):

Social isolation is defined in this study as a low frequency of contact with others; this definition discriminates social isolation from feelings of isolation or loneliness, in accordance with Townsend (Townsend, 1963).

Q 4 . I wasn’t completing sure on why logistic rather than linear regression was used in some cases. I.e. what was the value is coding WHO-5 scores as a binary variable over just looking at them as a linear scale. A sentence or two to explain and justify this approach might be useful to include in the ‘materials and methods’ section.

Answer: A review* demonstrated that the WHO-5 has high clinimetric validity. The review also demonstrated that it can be used as an outcome measure balancing the wanted and unwanted effects of treatments, is a sensitive and specific screening tool for depression, and its applicability across study fields is very high. We regard the cut-point of 12/13 as reasonable for this study**.

As a standard cut-off criterion of a total score is 12/13, less than 13 was regarded as a poor condition.

We revised the text accordingly. The text now adds (Line 105–Line 107 in the revised text): “WHO-5 is a sensitive and specific depression screening tool with very high applicability across study fields (Topp et al., 2015).” Also, the text now adds (Line 110–Line 111 in the revised text: “We coded WHO-5 scores as a binary variable with a standard cut-off criterion of 12/13. That is, scores of 12 or less were regarded as a poor condition.”

REFERNCE

* Topp, C.; Østergaard, S.; Søndergaard, S.; Bech, P. The WHO-5 Well-Being Index: A systematic review of the literature. Psychother Psychosom 2015, 84, 167–76. Doi: 10.1159/000376585.

** Awata, S.; Bech, P.; Koizumi, Y.; Seki, T.; Kuriyama, S.; Hozawa, A.; Ohmori, K.; Nakaya, N.; Matsuoka, H.; Tsuji, I. Validity and utility of the Japanese version of the WHO-Five Well-Being Index in the context of detecting suicidal ideation in elderly community residents. Int Psychogeriatr 2007, 19(1), 77–88. PMID: 16970832.

Q5 . There were some points in the discussion where I wasn’t clear if you were referring to your own findings or that of previous research. For example, on line 230 you say “The trajectories of social contact indicated differences between contact with family members and non-family members; the frequency of social contact with family or relatives increased, while that with friends or neighbors decreased.” But I couldn’t recall you talking about this finding in the results section…

Answer: I appreciated your comment, I deleted the sentence “The trajectories of social contact indicated differences between contact with family members and non-family members; the frequency of social contact with family or relatives increased, while that with friends or neighbors decreased.” in the revised text.

Q6.  Line 253, you say “influence of SNS…” but I don’t think you had told the reader prior to this what SNS referred to?

Answer: Although we measured the frequency of NFFC—which included SNS as well as via telephone, email, letters, and so on—with non-residential family and relatives, friends, and neighbors, we did not ask about the type and usage of NFFC in detail. Therefore, we revised the text as follows: “This difference in the present study may be caused by the influence of NFFC tools, which might make NFFC less effective for mental health compared to FFC” (Line 262–Line 264 in the revised text).

Furthermore, the text now reads: “First, the present study did not discuss SNS type and usage in detail, even though NFFC includes SNS as well as telephone, email, letters, and so on” (Line 277–Line 278 in the revised text).

Q7. I did not see the value in having the ‘Measures and analysis strategy’ heading within the discussion as most of the content that came under it was related to the results of the research.

Answer: I am very sorry for my misprint. We removed the ‘Measures and analysis strategy’ heading. We revised it to ‘Cross-sectional and longitudinal association of social contact with mental health’ (Line 251 in the revised text).